# Towards design of drugs and delivery systems with the Martini coarse-grained model

Lisbeth R. Kjølbye[1]† , Gilberto P. Pereira[1]† , Alessio Bartocci[2] ,
Martina Pannuzzo[3] , Simone Albani[4,5] , Alessandro Marchetto[4,5] ,
Brian Jiménez-García[6] , Juliette Martin[1] , Giulia Rossetti[4,7,8] ,
Marco Cecchini[2] , Sangwook Wu[3,9] , Luca Monticelli[1] and
Paulo C. T. Souza[1]*

[1]Molecular Microbiology and Structural Biochemistry (MMSB, UMR 5086), CNRS & University of Lyon, Lyon, France; [2]Institut de Chimie de Strasbourg, UMR 7177 CNRS, Université de Strasbourg, Strasbourg Cedex, France; [3]PharmCADD, Busan, South Korea; [4]Computational Biomedicine, Institute of Advanced Simulation (IAS-5) and Institute of Neuroscience and Medicine (INM-9), Forschungszentrum Jülich GmbH, Jülich, Germany; [5]Department of Biology, Faculty of Mathematics, Computer Science and Natural Sciences, RWTH Aachen University, Aachen, Germany; [6]Zymvol Biomodeling, Barcelona, Spain; [7]Jülich Supercomputing Centre (JSC), Forschungszentrum Jülich GmbH, Jülich, Germany; [8]Department of Neurology, Faculty of Medicine, RWTH Aachen University, Aachen, Germany and [9]Department of Physics, Pukyong National University, Busan, Republic of Korea

## Perspective

**Key words:**
coarse-grained models; molecular dynamics; Martini; drug design; drug delivery; cryptic pockets; transmembrane proteins; protein-protein interactions; soft delivery systems; PROTACS; lipid nanoparticles

**Author for correspondence:**
*Paulo C. T. Souza,
E-mail: paulo.telles-de-souza@ibcp.fr

†L.R.K. and G.P.P. contributed equally to this work.

### Abstract

Coarse-grained (CG) modelling with the Martini force field has come of age. By combining a variety of bead types and sizes with a new mapping approach, the newest version of the model is able to accurately simulate large biomolecular complexes at millisecond timescales. In this perspective, we discuss possible applications of the Martini 3 model in drug discovery and development pipelines and highlight areas for future development. Owing to its high simulation efficiency and extended chemical space, Martini 3 has great potential in the area of drug design and delivery. However, several aspects of the model should be improved before Martini 3 CG simulations can be routinely employed in academic and industrial settings. These include the development of automatic parameterisation protocols for a variety of molecule types, the improvement of backmapping procedures, the description of protein flexibility and the development of methodologies enabling efficient sampling. We illustrate our view with examples on key areas where Martini could give important contributions such as drugs targeting membrane proteins, cryptic pockets and protein–protein interactions and the development of soft drug delivery systems.

## Introduction

Recent studies have shown that the cost of drug discovery and development is, on average, higher than several hundred million dollars (Mohs and Greig, 2017; Schlander *et al.*, 2021). Moreover, several diseases such as Alzheimer, cancer, viral infections and cardiovascular diseases remain orphan of an effective, long-term and safe therapeutic protocol (Falzone *et al.*, 2018; Nishiga *et al.*, 2020; Brown and Wobst, 2021; Esang and Gupta, 2021). Current challenges in the development of novel therapeutic approaches include the unavailability of druggable binding pockets in the target structure (Weerakoon *et al.*, 2022) and the lack of effective delivery systems, which can improve drug pharmacodynamics (Wang *et al.*, 2021b).

Computational methodologies can speed up the drug discovery pipeline, decrease the associated costs and provide insight into the interactions between drugs and their targets, which is critical for rational drug design (Sliwoski *et al.*, 2014; Lin *et al.*, 2020). Computer modelling permeates both hit-identification and lead-optimisation stages of drug discovery pipelines. Computational methods have been used to predict protein–ligand binding modes (Śledź and Caflisch, 2018), binding affinities (Montalvo-Acosta and Cecchini, 2016), brain–blood barrier permeation (Crivori *et al.*, 2000), compound activity against a given target (Pereira *et al.*, 2018) or to identify and map potential binding sites (Yu and MacKerell, 2017; MacKerell *et al.*, 2020). Some of these methods rely on atomistic molecular dynamics (MD) simulations to produce configurational ensembles (Siebenmorgen and Zacharias, 2020). However, converging on sampling the potential energy landscape of large biomolecular complexes is challenging and limits the application of atomistic MD to smaller systems. Nonetheless, numerical simulations and docking studies can still contribute to studies of protein–ligand interactions or identification of hit compounds (Jorgensen, 2009; Bollini *et al.*, 2011; Frey *et al.*, 2013). Alternatively, purpose-built hardware and software can help simulate large systems at atomistic resolution as shown by the DESRES team (Dror *et al.*, 2011; Shaw *et al.*, 2021).

Drug delivery has also seen an increase in usage of computational modelling, mainly because current development pipelines rest upon unpredictable trial and error experiments. Molecular modelling offers an attractive platform for understanding and optimising delivery systems in a biologically relevant context (Wang *et al.*, 2021b). In this field, the limitations associated with system size and complexity are magnified. Several model systems exploring interaction with lipid bilayers have been constructed, with more realistic models mainly being of solid nanoparticles (NPs) such as gold NPs (Franco-Ulloa *et al.*, 2021; Salassi *et al.*, 2021). Limited studies have explored softer delivery systems like lipid-based NPs, mainly due to the lack of well-established computational protocols for constructing and studying these systems.

Coarse-grained (CG) modelling techniques alleviate sampling limitations of atomistic MD. The most widely used CG force field (FF) is the Martini FF (Marrink *et al.*, 2007). The newly developed Martini 3 (*Souza et al., 2021a*) improves sampling efficiency by merging together two to four non-hydrogen atoms and corresponding associated hydrogens into one interaction bead, with the bonded and non-bonded parameters derived from a combination of bottom-up and top-down approaches, respectively. In parallel with the development of the Martini 3 FF, other CG approaches were pursued. For instance, some recent developments in protein CG models include the SIRAH2.0 FF (Machado *et al.*, 2019), SPICA (Kawamoto *et al.*, 2022) or the recently developed ProMPT, an alternative polarisable Martini model (Sahoo *et al.*, 2022).

The Martini 2 FF currently supports a wide array of parameters for proteins, different lipid types, polymers, DNA and RNA (Monticelli *et al.*, 2008; López *et al.*, 2009; De Jong *et al.*, 2013; Uusitalo *et al.*, 2015, 2017; Grünewald *et al.*, 2018; Salassi *et al.*, 2018). Four main bead types were developed based on the polarity of chemical groups. These particles are further subdivided depending, for example, on their hydrogen-bonding capabilities (Marrink *et al.*, 2007). Limitations of the Martini 2 model included overstabilisation of some biomolecular interactions, mainly noted for proteins and sugars (Alessandri *et al.*, 2019) and the narrow range of chemical groups represented by the available beads (Kanekal and Bereau, 2019). The new version 3 (*Souza et al., 2021a*) addressed these issues and now provides promising solutions for drug design and delivery. New Martini 3 CG models allow simulations of more complex systems, facilitating the study of important biomolecular processes like ligand binding (Souza *et al.*, 2020), fusion events (Bruininks *et al.*, 2020), and the distribution of drugs within particle or carrier delivery systems (Casalini, 2021). This enables understanding of the forces behind encapsulation and drug release, which furthers the optimisation and development of delivery systems, as well as the interactions, which drive ligand binding, fundamental for drug design campaigns.

The Martini 2 and 3 FFs have been applied to study different biomolecular systems, among them proteins, membranes or vesicles, and is increasingly being used in the field of materials sciences (Marrink *et al.*, 2019; Alessandri *et al.*, 2021; Marrink *et al.*, 2022). Examples exist of CG simulations studying fusion of delivery systems, such as lipoplexes or nanoemulsions, with lipid bilayers (Lee *et al.*, 2012; Bruininks *et al.*, 2020; Gupta *et al.*, 2021; Machado *et al.*, 2022). In 2020, Bruininks *et al.* (2020) used CG modelling to simulate the fusion of a cationic lipoplex containing DNA with a simple membrane model representing the endosomal membrane. This is one of the first stepping stones for using CG models to explore nucleic acid (NA) release. For drug binding, the potential of CG-Martini simulations in studies of protein–ligand binding was shown in the work of Negami *et al.* (2014) where they studied

protein–ligand binding for two systems, levansucrase-glucose and LinB-1,2-dichloroethane. A more recent example is the application of Martini 3 FF to study protein–ligand binding in T4 Lysozyme with different small molecules and several pharmacologically relevant targets, such as G protein-coupled receptors (GPCRs), kinases and one example of nuclear receptor (Souza *et al.*, 2020), achieving quantitative agreement with experimental binding affinities. Other examples are present in the literature (Delort *et al.*, 2017; Ferré *et al.*, 2019; Jiang and Zhang, 2019; Dandekar and Mondal, 2020; Negami *et al.*, 2020). Furthermore, the application of Gō models (Poma *et al.*, 2017) in the Martini 3 model leads to an improved description of protein flexibility while preserving computational efficiency. Combined with the new Martini 3 small molecule library (*Souza et al., 2021b*; Alessandri *et al.*, 2022), CG Martini models now gather the conditions for successful applications in structure-based drug discovery campaigns.

In this perspective, we discuss potential applications of Martini 3 CG simulations to topics relevant for drug discovery and development pipelines, including design of innovative therapies, binding site identification and optimisation of soft delivery systems.

## Protein conformation and cryptic pockets

Drug-binding sites are usually pockets or grooves located on the surface of the target protein (Vajda *et al.*, 2018) accessible even in the absence of the drug (Vajda *et al.*, 2018). However, since proteins are dynamic objects, 'hidden' binding sites may appear in the presence of an interacting compound (Oleinikovas *et al.*, 2016; Vajda *et al.*, 2018). These cryptic pockets are often not apparent on the unbound protein surface, only transiently opening up as rare events or shaping themselves in the presence of a ligand (Fig. 1). Cryptic sites can provide unforeseen tractable drug target sites, thus expanding the druggable proteome considerably (Vajda *et al.*, 2018; Hopkins and Groom, 2002). On one hand, cryptic pockets offer the prospect to design allosteric drugs (Wenthur *et al.*, 2014), a strategy that could be exploited as a therapeutic path towards treating cancer (Zhong *et al.*, 2021), diabetes (Wang *et al.*, 2021a), and more recently, SARS-CoV-2 infections (Zimmerman *et al.*, 2021). On the other hand, cryptic pockets commonly occur at protein–protein interfaces [PPI; (see section 'Drugs targeting protein–protein interactions')]. Therefore, the ability to discover and target cryptic pockets would also enable the design of compounds targeting PPIs en route to new therapeutic formulations (Wells and McClendon, 2007; Shan *et al.*, 2022).

Several approaches have been proposed for the identification of cryptic sites. While some of them are entirely based on the analysis of protein crystallographic structures (Le Guilloux *et al.*, 2009), the majority use MD for the identification of cryptic sites (Le Guilloux *et al.*, 2009; Kokh *et al.*, 2013; Laurent *et al.*, 2015; Cimermancic *et al.*, 2016; Kuzmanic *et al.*, 2020; Zheng, 2021; Shan *et al.*, 2022).

Cryptic sites are not usually captured in the 180,000+ tridimensional structures obtained by state-of-the-art experimental methods (Bank, 2021); their opening generally occurs on the microsecond-to-millisecond time timescale (Kuzmanic *et al.*, 2020). These timescales are only accessible to all-atom (AA) MD simulations relying on specialised hardware, like the Anton3 (Dror *et al.*, 2011; Shaw *et al.*, 2021), or massive distributed computing, as in the Folding@home project (Zimmerman *et al.*, 2021), but not yet for standard GPU-accelerated hardware (Schlick and Portillo-Ledesma, 2021). As a workaround, AA MD-based approaches involve the addition of hydrophilic (e.g. acetic acid, isopropanol) or hydrophobic (e.g. benzene) molecules in the simulation, the

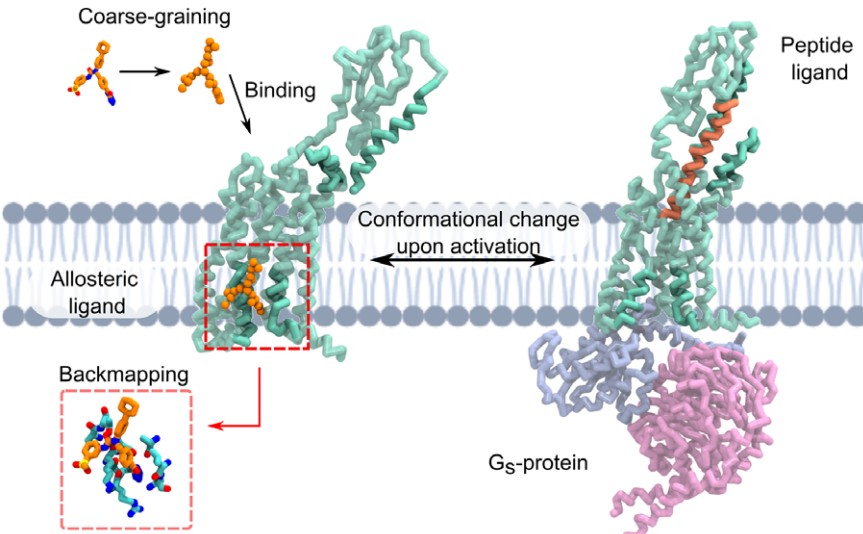

**Fig. 1.** Schematic representation of a GPCR (PDB IDs 5XEZ & 6LMK) in inactive (left) and active (right) conformations with an allosteric and peptide ligand bound, respectively. Large conformational changes occur upon binding of the peptide ligand and G$_s$-protein binding intracellularly, which represent possible dynamics that could be observed with Martin combined with Gō-models. The allosteric pocket in the transmembrane domain exemplifies the possibility to use Martini models for identifying transmembrane pockets, allosteric or cryptic, in various complex membrane compositions. Once a ligand is bound, backmapping is a possibility to obtain higher resolution information for further ligand optimisation or design. All figures were rendered using VMD (Humphrey *et al.*, 1996).

so-called mixed-solvent MD (Ghanakota and Carlson, 2016), the addition of the drug in high concentration, the so-called 'flooding' MD approach (Amaro and Li, 2010; Gray *et al.*, 2017), or fragment-based screening (MacKerell *et al.*, 2020). However, also in this case the opening of cryptic pockets may still require several microseconds (Kuzmanic *et al.*, 2020). Enhanced sampling approaches have also been used (Bono *et al.*, 2013; Herbert *et al.*, 2013; Oleinikovas *et al.*, 2016). For the collective variable (CV)-based approaches, the central challenge is choosing a suitable CV (Kuzmanic *et al.*, 2020). For the CV-independent methods, running many simulations still entails high computational costs (Earl and Deem, 2005; Kokh *et al.*, 2016), which constitutes the main limiting step.

The Martini 3 CG FF, with its increased accuracy and expanded coverage of the chemical space (*Souza et al., 2021a*; Alessandri *et al.*, 2022), represents a competitive alternative for extracting and targeting druggable structures on such timescales and/or predicting ligand–target interactions (Souza *et al.*, 2020, 2021*b*). So far, one of the limitations of the Martini models is the description of proteins' conformational flexibility (Poma *et al.*, 2017; Souza *et al.*, 2019), which is fundamentally linked to biological function (Henzler-Wildman and Kern, 2007; Luo, 2012; Veesler and Johnson, 2012; Campaner *et al.*, 2017; Hadden *et al.*, 2018; Matthes *et al.*, 2018; Maggi *et al.*, 2020; Bolnykh *et al.*, 2021; Noreng *et al.*, 2021; Jackson *et al.*, 2022) and pivotal to design new therapeutics (Hammes, 2002; Campaner *et al.*, 2017; Sengupta and Udgaonkar, 2019; Schulz-Schaeffer *et al.*, 2020; Gossen *et al.*, 2021; Xiao *et al.*, 2021; Zhao *et al.*, 2021; Margreiter *et al.*, 2022).

Commonly, Martini-based approaches implement an elastic network (EN), that is addition of a network of harmonic restraints to stabilise protein tertiary structure (Periole *et al.*, 2009). The restraints are usually added based on a distance criterion, introducing a strong bias towards the starting conformation (Periole *et al.*, 2009). Different strategies were devised to address this issue (Deplazes *et al.*, 2012; Lelimousin *et al.*, 2016; Poma *et al.*, 2017): *(i) localised distance-restraints on selected secondary structure elements*, often driven by experimental information. For example, this approach was used to study the activation of the epidermal growth

factor receptor, coupling Martini 2 CG simulations with enhanced sampling techniques, such as well-tempered metadynamics (Barducci *et al.*, 2008) and distance-based restraints on transmembrane helices (Lelimousin *et al.*, 2016). *(ii) implementation of G-ō-like* models (GōMartini) by establishing a Lennard–Jones (LJ) potential based on the contact map of the native protein structure instead of the harmonic based potential (Poma *et al.*, 2017; Souza *et al.*, 2019; Mahmood *et al.*, 2021). Different contact map definitions for GōMartini were tested on three protein systems (cohesin, titin and ubiquitin) and reproduced protein flexibility as observed in AA simulations. Two different contact map definitions were tested and compared against ENs. The first variant only considers van der Waals spheres (OV) overlaps, while the second builds on top of the OV approach and includes chemical information from the atoms in question. Here a contact between residues requires that the number of attractive contacts between atoms be larger than the number of repulsive ones (Wołek *et al.*, 2015; Poma *et al.*, 2017).

Other approaches also exist to characterise conformational transitions between two or more conformational states in the presence of ENs (Kim *et al.*, 2002; Miyashita *et al.*, 2003; Feng *et al.*, 2009; Das *et al.*, 2014), like gradually switching between two different types of EN connectivity through a switching parameter or the so-called 'generalised elastic network' (Poma *et al.*, 2018) which, within a given cut-off, implements a canonical EN with harmonic potentials and above the chosen cutoff instead implements Gō-like contacts to the system. Recently, adaptive ENs have been developed as well (Kanada *et al.*, 2022). These strategies represent a potential powerful development strategy for Martini models.

The Martini model can be coupled with strategies to introduce proteins' dynamics, as discussed above, and combined with enhanced sampling approaches to ideally push the system towards the exploration of 'rare' events, like cryptic pocket opening. The use of artificial intelligence algorithms to identify and speed-up the slower modes, as done in Bonati *et al.* (2020 and 2021) could also be exploited to steer cryptic pockets' opening. This represents a valid and computationally cheaper solution to identify and target cryptic

pockets. Once possible pockets are identified at the CG level, the protein structure could be converted into atomistic resolution (Wassenaar *et al.*, 2014; Vickery and Stansfeld, 2021) for further investigation and ligand design within a virtual screening (VS) workflow.

## Protein binding pockets in membrane environments

A fast-growing area for Martini simulations is the analysis of protein–ligand interactions in membrane environments, which is experimentally and computationally rather challenging. Here, the ligand may be an endogenous lipid, that is, natively part of the physiological environment, or an exogenous compound targeting an allosteric pocket of a transmembrane protein or at the protein–lipid interface (Fig. 1). In this context, the structural characterisation of the ligand-binding sites in the transmembrane region and ranking based on binding energetics extracted by CG MD simulations is particularly attractive. This is even more so because standard computational approaches for protein–ligand binding like molecular docking, that do not account for the specificity of the membrane environment, that is, the strong hydrophobic character and the competition with native lipids, are prone to fail. Recently, several CG MD investigations of protein–ligand interactions in the transmembrane region of pharmacologically relevant targets have been reported. In general, the common computational strategy involves: *(i) binding-site identification* and structural characterisation of the protein–ligand complex using, among other methods, unbiased CG MD simulations and ligand-density maps (Ferraro *et al.*, 2016; Dämgen and Biggin, 2021); *(ii) ranking of binding modes* by binding affinity calculations based on equilibrium MD (Souza *et al.*, 2020), potential of mean force (PMF), alchemical transformations, metadynamics (Corey *et al.*, 2019) or binding saturation curves (Ansell *et al.*, 2021); and *(iii) structural refinement* of the protein ligand complex via backmapping to atomistic models (Wassenaar *et al.*, 2014). Overall, the main advantage of CG modelling is the ability to converge on sampling the protein–ligand conformational space, currently out-of-reach by typical unbiased atomistic simulations. As a result, within the limits of the accuracy of the model, trends in dissociation constants ($K_d$) and rates ($K_{off}$) can potentially be accessed from unbiased MD (Souza *et al.*, 2020, 2021*b*).

The vast majority of CG MD analyses of protein–ligand interactions in membrane environments involve protein–lipid binding based on Martini 2.2 simulations (De Jong *et al.*, 2013). The use of the Martini 2.2 FF has allowed not only to discern specific versus nonspecific interactions but also to characterise the energetics involved in the binding reaction. Earlier efforts focused on the prediction of the binding site(s) for cholesterol, which is the most abundant endogenous steroid in mammalian cell membranes and was shown to modulate several membrane proteins including ion channels. Using multi-microsecond CG MD simulations of a homology model of the serotonin transporter embedded in a raft-like membrane, Ferraro *et al* (2016) provided evidence of the existence of specific binding sites for cholesterol, identifying a hotspot that largely overlaps with the cholesterol-binding site illuminated by X-ray crystallography of the closely related dopamine transporter (Ferraro *et al.*, 2016). By combining CG MD simulations and PMF calculations, Ansell *et al* (2021) characterised the interaction between cholesterol and several membrane proteins including an ATP-dependent pump, a sterol receptor/transporter protein and a member of the TRP ion-channel family. A similar analysis of the

chemokine receptor 3, a GPCR responsible for trafficking white blood cells, allowed for the identification of six cholesterol-binding sites, suggesting that recognition of cholesterol at these sites may modulate the affinity for agonists/antagonists allosterically via a rigidification of the protein structure (van Aalst *et al.*, 2021). Using CG MD simulations and lipid-density maps, Damgen and Biggin (2021) explored the affinity of cholesterol and different lipid types for the glycine receptor channel in its active and resting states and found that lipids may act as allosteric modulators because their strength of binding strongly depend(s) on the physiological state of the receptor. In a similar study, protein–lipid interactions on the homologous nicotinic acetylcholine receptor were investigated using a complex quasi-neuronal membrane composed of 36 species of lipids, including cholesterol, in a binding competition assay (Sharp and Brannigan, 2021). Interestingly, the CG MD simulations suggested that cholesterol binds to concave inter-subunit sites and polyunsaturated fatty acids prefer convex sites at the outer transmembrane helix M4, while monounsaturated and saturated lipids are enriched at the protein–lipid interface (Sharp and Brannigan, 2021). Recently, the interaction of the anionic lipids cardiolipins with 42 inner membrane proteins from *Escherichia. coli* has been investigated by CG MD simulations. Overall, >700 independent cardiolipin binding sites were identified and structurally characterised, thus providing a molecular basis for protein–cardiolipin interactions (Corey *et al.*, 2021). In the context of systematic comparative analyses, the method by Ansell *et al.* (2021) for protein–ligand binding affinities based on binding saturation curves appears particularly appealing as a high-throughput approach for binding-site comparison and ranking.

In addition to protein–lipid interactions, a potential area of development for CG simulations involves the exploration of modulatory ligand binding, such as agonists, antagonists and allosteric modulators, to the transmembrane region of proteins. In this case, and unlike for most lipid molecules, a serious difficulty is introduced by the lack of off-the-shelf CG parameters to model the ligand(s). As a result, examples of studies focusing on the allosteric modulation of transmembrane proteins via protein–ligand interactions are still rare in the literature. One of them focused on the investigation of the binding pathway of two orthosteric agonists of the μ-opioid receptor, that is, fentanyl and morphine (Sutcliffe *et al.*, 2021). Using CG MD simulations and free energy calculations, Sutcliffe *et al.* (2021) compared the aqueous and lipophilic binding pathways to the orthosteric site and found that the synthetic opioid fentanyl prefers the lipophilic route, which might explain its lower susceptibility to overdose reversal. Since more and more high-resolution structures of relevant pharmacological targets highlight the existence of multiple allosteric sites in the transmembrane region of these proteins (Cerdan *et al.*, 2020), the development of automatic parameterisation tools to facilitate the setup of CG MD simulations, similar to what is currently available for AA MD, is expected to leverage more exploratory analyses of protein–ligand interactions in the membrane environment and open to high-throughput screening powered by CG MD simulations. Additionally, the new Martini 3 FF offers an extended chemical space (Souza *et al.*, 2020, 2021*a*, 2021*b*), providing an excellent platform for developing automatic parameterisation tools for ligands.

## Drugs targeting protein–protein interactions

PPIs have been considered as promising drug targets since the early 2000s, with the hope to overcome the decline in the efficiency of

conventional drug development. Three major types of PPI modulators currently described in the literature are small molecules, antibodies and peptides (Mabonga and Kappo, 2019; Lu et al., 2020; Martino et al., 2021). Small molecules typically require a prototypical binding site. The PPI interface is usually flat, shallow and hydrophobic, without an actual pocket where small-molecule ligands may bind (Lu et al., 2020). The natural alternative would be to increase the size of the modulator to maximise PPI interface coverage and establish many hydrophobic contacts (Lu et al., 2020). However, increasing small-molecule-based PPI modulator size may lead to undesirable pharmacokinetic profiles (An and Fu, 2018; Lu et al., 2020; Martino et al., 2021). Antibodies present an alternative therapeutic avenue, since these can fully cover the PPI interface due to their size (Bojadzic and Buchwald, 2018; Martino et al., 2021) and there is potential for the general application of antibody-based therapies when combined with novel drug delivery systems (Slastnikova et al., 2018). Peptides can also be used to modulate PPIs as they bind the PPI interface with high affinity (Cabri et al., 2021), but they may exhibit short half-lives and toxicity risks (Gupta et al., 2013; Nevola and Giralt, 2015; Mabonga and Kappo, 2019). Examples of small-molecule PPI modulators are venetoclax, to treat chronic lymphoblastic leukaemia (Lu et al., 2020), and pomalidomide to treat myeloma (Dimopoulos et al., 2014). ALRN-6924 is an α-helical peptide aimed at leukaemia therapy (Carvajal et al., 2018) while Bavencio is an antibody-based drug targeting Merkel cell carcinoma (Boyerinas et al., 2015).

Recently, a new type of PPI modulator technology gained momentum: the Proteolysis Targeting Chimeras (PROTACs) (Sakamoto et al., 2001). These bivalent molecules consist of a linker connecting a small molecule binding the target (i.e. 'warhead') and a second small molecule binding an E3 ligase (the 'recruiter'), acting as a PPI enhancer like molecular glues (Wang et al., 2020; Alabi and Crews, 2021; Bond and Crews, 2021; Békés et al., 2022). Simultaneous binding of both proteins by the PROTAC brings them into proximity, provoking target ubiquitination and posterior degradation by proteasome machinery (Wang et al., 2020; Alabi and Crews, 2021; Bond and Crews, 2021; Békés et al., 2022). Compared to small-molecule inhibitors, PROTACs work catalytically, requiring less compound concentration, having fewer off-target effects and exhibiting improved target selectivity (Troup et al., 2020; Alabi and Crews, 2021; Békés et al., 2022). In the last years, PROTACs attracted the interest of academic and pharmaceutical companies and currently two molecules developed by Arvinas were forwarded to Phase II clinical trials (Petrylak et al., 2020; Békés et al., 2022). Key steps in PROTAC development include the selection of the E3 ligase to pair with the target of interest (Cecchini et al., 2021), the accurate prediction of the ternary complex structure (Zaidman et al., 2020) and linker design (Troup et al., 2020; Bemis et al., 2021).

Computational modelling and simulations can help the rational design of PROTACs (Fig. 2). In the absence of ternary complex crystal structures, which must contain the target, the PROTAC and the ligase, one of the first steps of in silico design is sampling the conformational landscape of the complex, which is achievable by protein–protein docking (Hayashi et al., 2018; Drummond and Williams, 2019; Drummond et al., 2020; Rosell and Fernández-Recio, 2020; Zaidman et al., 2020; Bluntzer et al., 2021; Bai et al., 2021, 2022; Weng et al., 2021) and/or MD simulations at various levels of detail (Rakers et al., 2015; Yu et al., 2019; Perez et al., 2021). For very large systems however, AA MD can become prohibitively expensive (Durrant and McCammon, 2011; Amaro et al., 2018; Jung et al., 2021). This is particularly true when considering a VS campaign applied to ternary complexes in explicit solvent, due to

system size and complexity. As an alternative, docking and MD simulations based on the Martini 2 and 3 CG framework (Roel-Touris et al., 2019; Roel-Touris and Bonvin, 2020; Souza et al., 2021b) may be used to facilitate the study of these large macromolecular systems. In a first stage, CG protein–protein docking can be used to capture the most important features of the interaction complex, providing many potential binding modes. It can then be combined with long and affordable CG MD simulations to probe complex stability, which is critical for PPI drug discovery. One example of CG-docking methods is HADDOCK (Roel-Touris et al., 2019; Roel-Touris and Bonvin, 2020). A limitation of some docking approaches is the treatment of proteins as rigid bodies (Vakser, 2020; Harmalkar and Gray, 2021). Recently, docking approaches including protein flexibility have been developed, including 'divide-and-conquer' (Karaca and Bonvin, 2011) and normal mode analysis-based strategies (May and Zacharias, 2008; Moal and Bates, 2010; Jiménez-García et al., 2018; Diaz et al., 2021). Alternatively, GōMartini simulations (Poma et al., 2017) could be used to cheaply produce protein–protein conformations which, after a back-mapping procedure, could be used in ensemble docking (Amaro et al., 2018).

Massive protein–protein docking for target identification (Zhang et al., 2014) of other PPI modulators can also greatly benefit from the use of CG approaches. In the case of PROTACS, not only the target but also the choice of E3 Ligase is fundamental for the stability of the ternary complex and cell-specific target degradation (Békés et al., 2022). Only a limited number of E3 ligases have been explored towards PROTAC development (Burslem and Crews, 2020; Troup et al., 2020; Alabi and Crews, 2021). Examples are the von Hippel-Landau or Cereblon E3 ligases (He et al., 2020; Bricelj et al., 2021). However, some ligases, for which there is currently no crystal PROTAC ternary complex available, are known to be enriched in specific cell types (Békés et al., 2022). Combining CG docking between a PROTAC-containing target and several candidate ligases separately with subsequent CG MD simulations could help to identify the most suitable target-ligase pair, enabling cell-type-based therapeutic PROTAC approaches.

Another PROTAC-specific challenge is the design of the linker portion (Alabi and Crews, 2021) as there exist no common practices or guidelines, and linker size and flexibility affect the degradation efficiency of PROTACs (Cyrus et al., 2011; Crew et al., 2018; Troup et al., 2020). Optimal linkers should be long and flexible enough to promote a ternary complex orientation that allow ubiquitin transfer to the lysines on the target surface. However, overly flexible linkers may hamper target degradation efficiency (Cecchini et al., 2021). CG-based approaches can help linker optimisation. For example, PROTAC CG docking simulations could be used to evaluate the possibility of other PROTAC molecules fitting into the available volume at the binding interface of a ligase/target complex. One route would be by harnessing structural data like the warhead-recruiter distances, extracted from ternary complexes from the Protein Data Bank (Burley et al., 2021) or from protein–protein docking experiments, as constraints. Filtering the predicted complexes using this information in combination with the docking score and other observables would allow retrieval of the best binding poses per system (Zaidman et al., 2020). From the most stable complexes, probed by CG MD, a linker template could then be designed and subsequently used in VS campaigns targeting chemically diverse linker libraries. Further, chemical modifications around the linker template would enable fine tuning of PROTAC properties like solubility, lipophilicity or toxicity effects (Troup et al., 2020). Recently,

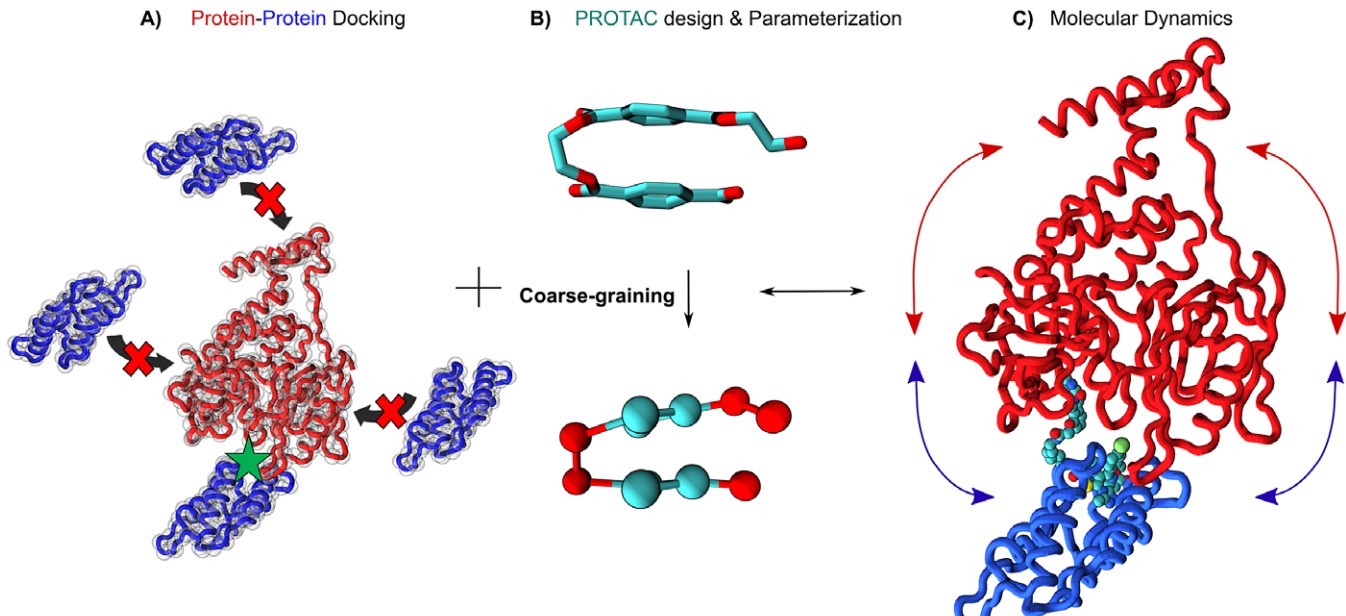

**Fig. 2.** Important steps in PROTAC design for drug discovery campaigns. (*a*) Protein–protein docking either at the atomistic (ribbons) or coarse-grained level (red and cyan spheres). The E3 ligase is represented in red and the target protein in blue. (*b*) Coarse-graining of a small -molecule using the Martini 3 force field. (*c*) Dynamical motions of the ligase and the target (blue and red arrows, respectively) are important to query ternary complex stability in the presence of the PROTAC (represented as van der Waals spheres). All figures were rendered using VMD (Humphrey *et al.*, 1996). The ternary complex structure is from Nowak *et al.* (2018) with the PDB ID code 6BN7.

the group of Kihlberg illustrated that PROTACs cell permeability is deeply related to the linkers' conformational flexibility. Although these compounds do not conform to oral bioavailability defined by the Lipinski rule-of-5 (Lipinski *et al.*, 2001), by acting as 'molecular chameleons' they are able to fold-in on themselves in aqueous solution and reduce their solvent-accessible polar surface area to increase cell permeability and then unfurl after crossing the membrane (Atilaw *et al.*, 2021). Thus, some of the key factors playing a role in PROTAC cell permeability are linker size (Klein *et al.*, 2020), polarity and rigidity (Atilaw *et al.*, 2021), further highlighting the importance of a rational linker design strategy. Similar concerns related to polarity and membrane permeability are also prevalent in PPI-targeting peptide design (Sugita *et al.*, 2021). As such, transfer free energy calculations carried out at the CG level could enable the direct investigation of the ability of different PPI modulators to cross biological membranes in an efficient and affordable manner while still achieving a high degree of accuracy.

## Tuning soft nanoparticles with Martini

Drug efficacy correlates with the ability of the drug to reach the target site in sufficient quantities. A high percentage of approved drugs display low aqueous solubility and are fast degraded. To tackle these problems, delivery systems have been developed (Malmsten, 2006; Wang *et al.*, 2021b). Different physicochemical properties of the delivery system, such as morphology, composition and stiffness, can contribute to the drug solubility, targeting efficiency and stability (Zhang *et al.*, 2015; Yu *et al.*, 2018). As drug carrier rigidity affects physiological membrane crossing, developing soft nanoparticle (SN) systems that can easily deform appears attractive.

SNs include carriers consisting of lipids, polymers or surfactants. Lipid-based carriers are generally biocompatible and highly permeable; however, they exhibit low mechanical stability

(Sercombe *et al.*, 2015). Polymer-based carriers, on the other hand, have higher mechanical stability but lower biocompatibility and permeability (Jana *et al.*, 2021). It is also possible to combine lipids and polymers and to harness the advantages of each component (Reimhult and Virk, 2021). Studies have shown that the mechanism of delivery for SNs depends on their morphology and composition, which is in turn correlated with the distribution of the drug within the carrier (El Maghraby *et al.*, 2008). However, little is known about the morphology and mechanism of delivery for these hybrid systems (Reimhult and Virk, 2021).

CG modelling is a valuable tool for investigating the formation of SNs, their morphology, drug distribution within the carrier and the mechanism of delivery, including the interaction with different biological membranes (Yang *et al.*, 2021; Parchekani *et al.*, 2022). The first obstacle for using CG models is constructing the system. Fortunately, an increasing number of tools have been developed for building such CG models, examples being TS2CG, Charmm-GUI and Nano Disc builder, allowing the construction of vesicles and other SNs for drug delivery (Qi *et al.*, 2015; Hsu *et al.*, 2017; Kjølbye *et al.*, 2020; Pezeshkian *et al.*, 2020), the Polyply package for constructing polymer-based systems (Grünewald *et al.*, 2022) or the *Insane.py* script for bilayers (Wassenaar *et al.*, 2015). In combination, protocols for simulating soft delivery systems have also started to appear in the literature (Bruininks *et al.*, 2019). Several CG studies have been performed using the Martini 2 model, investigating the morphology, size and internal organisation of the different components in lipid and polymer-based carriers (Hashemzadeh *et al.*, 2020; Bono *et al.*, 2021; Gao *et al.*, 2021). Among the first described SNs are liposomes, consisting of a lipid bilayer surrounding a hydrophilic core, capable of trapping both hydrophobic and hydrophilic drugs. Liposomes were the first delivery system to reach clinical application (Doxil) (James *et al.*, 1994) and have been widely used and characterised for many different therapeutics (Allen and Cullis, 2013). Further development of liposomes resulted in cationic lipids and subsequently

cationic polymers for delivery of NAs, which proved invaluable at the outbreak of the COVID-19 pandemic (Polack *et al.*, 2020; Baden *et al.*, 2021). Cationic lipids or polymers can condense NA efficiently, thanks to the electrostatic interaction with the negatively charged NA to form lipoplexes and polyplexes, respectively (Li and Szoka, 2007; Schlich *et al.*, 2021). Accurate description of the electrostatic interactions is a major challenge in the case of highly charged lipo- or polyplexes. The challenge could be tackled by developing polarisable Martini models, so far only available for water, ions and proteins with the Martini 2 FF (Yesylevskyy *et al.*, 2010; De Jong *et al.*, 2013; Michalowsky *et al.*, 2017, 2018; Sahoo *et al.*, 2022).

The main drawback of permanently charged cationic components is their toxicity and rapid elimination from circulation (Li and Szoka, 2007; Schlich *et al.*, 2021). To avoid the toxicity and increase the circulation time and stability, particles can be covered by a PEGylated lipid shield (Li and Szoka, 2007), although PEGylation has shown to diminish particle uptake in target cells (PEG dilemma) (Gjetting *et al.*, 2010). The Martini model has eased the way to study polymer coating in membranes (Grünewald *et al.*, 2018; Lemaalem *et al.*, 2020) and NPs (Pannuzzo *et al.*, 2020).

A step further in the optimisation led to ionisable components, resulting in the formulation of lipid nanoparticles (LNPs) (Schlich *et al.*, 2021) and dendrimers (Palmerston Mendes *et al.*, 2017), branched polymers with well-defined molecular weights. The ionisable components are positively charged at low pH to encapsulate NA, and neutral at higher pH, for example, in the blood, thereby avoiding the drawbacks of lipo- and polyplexes. However, it has been shown that only 2–3% of the nucleotide drug load reaches the cytosol using LNPs (Gilleron *et al.*, 2013). Once the LNP or dendrimer is endocytosed, endosomes eventually fuse with lysosomes and their cargo is degraded. For optimised release, the cargo needs to escape the endosome before fusion with the lysosome (Schlich *et al.*, 2021). A general understanding of the delivery mechanism and its *pH* dependence is lacking. For investigating *pH* dependent release routes or interactions with NA, constant *pH* CG approaches are available (Grünewald *et al.*, 2020; Aho *et al.*, 2022). As a proof of concept, collective interactions between titratable sites in a G5 dendrimer poly(propylene imine) were simulated at different *pH*

values, revealing how the particle expands in radius and increases in degree of protonation with decreasing *pH*, consistent with previous atomistic studies (Grünewald *et al.*, 2020).

The delivery depends on the structural properties of the carrier. For LNPs, two different internal organisations have been proposed based on CG modelling with Martini 2 and cryo-transmission electron microscopy (Leung *et al.*, 2012; Kulkarni *et al.*, 2018). Understanding the structure–activity relationship is of paramount importance for the rational design of optimised LNPs and for cell-specific targeting. Cell specific LNPs can be built by changing one or more of the lipid components (Liu *et al.*, 2021; Žak and Zangi, 2021), but models of synthetic lipids are not always available. The combination of the Martini 3 FF, with its extended chemical space, and building tools enables the prediction of properties of both empty structures of SNs, such as LNPs, and complexes with cargo of various sizes, from small interfering RNA (siRNAs) to large messenger RNA (mRNA) molecules, enabling studies of the internal organisation and interactions.

Cell specificity and drug efficacy can, in principle, be optimised in terms of interaction and fusion with the endosomal membrane. To this end, Martini 2 models of complex membranes have previously been constructed (Ingólfsson *et al.*, 2014, 2020), which demonstrates the possibility of studying the interaction between various SNs formulations (Lee *et al.*, 2021) and cell-specific plasma and endosomal membranes. However, this field remains to be explored.

In perspective, there is a general need to implement alternative design strategies to further optimise SNs. Tuning the chemical groups of the lipids or polymer, along with ratios of components, is key for altering the properties of SNs. However, synthesising and testing combinations of lipid or polymer variables are costly and time consuming. *In silico* screening of promising formulations is a viable alternative to study the role that each component plays in the SN morphology and delivery process (Fig. 3). One drawback when studying lipid-based systems using CG approaches is the loss of resolution compared to AA representations. For instance, changes in tail composition are not always well captured. Nevertheless, in the new version of Martini different tail chemistries can be easier to represent in the future as a result of the use of small and tiny beads.

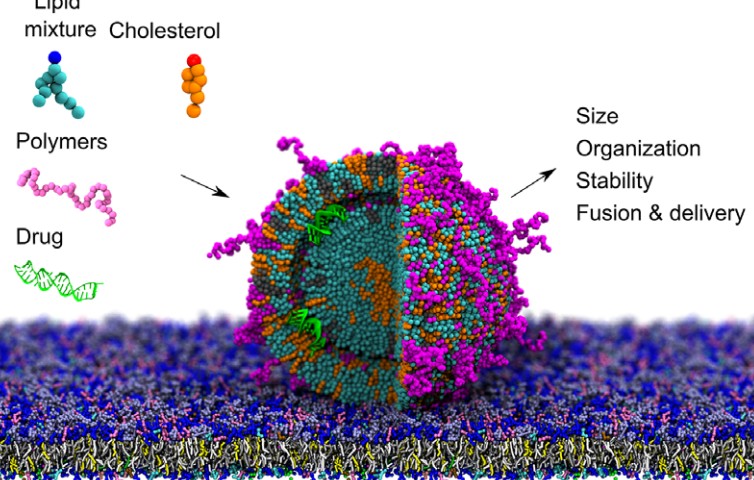

**Fig. 3.** CG modelling enables predictions of organisation, size and stability of SNs containing various building blocks and cargo. Moreover, it can be used to study the interaction between various SN formulations and biological barriers, such as plasma and endosomal membranes. All figures were rendered using VMD (Humphrey *et al.*, 1996).

## Summary and future directions

For *in silico* studies of large complex systems, aiming at identifying possible druggable sites, predicting and optimising protein ligand binding for drug design or studying drug delivery systems, the Martini model provides an efficient approach relative to AA MD. Due to the timescales reachable by CG Martini simulations, it is possible to probe systems with respect to pockets formed transiently, interesting for drug discovery campaigns. However, further benchmarking is required to assess the accuracy of the FF. Furthermore, maintaining proteins' tertiary structure in Martini models requires inclusion of EN or GōMartini potentials. Thus, a reasonable definition of the contact map, from which to draw the network of potentials, is critical. Improvements to the definition of contact maps may consider an ensemble of conformations and use knowledge of hydrogen bonds and residue protonation. Additionally, the use of LJ potentials in GōMartini enables the development of multi-basin models, as previously shown for AA MD (Okazaki *et al.*, 2006), combining Gō-models of different protein conformations to promote conformational transitions.

Currently, a major challenge towards the use of the Martini CG model in drug design and drug delivery is the automatic parameterisation of ligands and components of delivery vectors, including AA-to-CG mapping, construction of the bonded parameters and bead-type assignment. To address this issue, tools like Swarm-CG (Empereur-Mot *et al.*, 2020) or PyCGTOOL (Graham *et al.*, 2017) have been developed. However, these approaches focus solely on optimising the bonded parameters. Automated parameterisation workflows for Martini 2 models of small molecules are available (Bereau and Kremer, 2015; Potter *et al.*, 2021), including mapping, bonded-parameter definition and bead type selection based on optimisation of oil–water partitioning free energies. However, since the covered chemical space is larger in Martini 3, adapting these codes to Martini 3 is not straightforward. Equally important will be the generation of curated and extended libraries of Martini models, such as MAD (the MArtini Database server – https://mad.ibcp.fr), which can be used as reference to access the accuracy of such automatic approaches (Hilpert *et al.*, 2022). However, the current library of Martini 3 small-molecule models (Alessandri *et al.*, 2022) may already allow initial benchmarks based on fragment-based strategies. Another challenge is the backmapping procedure from CG to AA resolution (Wassenaar *et al.*, 2014; Vickery and Stansfeld, 2021), as protein side-chain directionality is kept, but the binding mode may not be accurate. A standard solution is to perform cycles of energy minimisation and equilibration on the backmapped structure to improve side-chain packing. Another option in this direction would involve the use of machine learning methods to optimise side-chain orientation (Misiura *et al.*, 2022).

Backmapping and small-molecule automatic parameterisation are fundamental goals towards VS of molecules targeting PPI systems, like PROTACS. Additionally, available tools for CG protein–protein docking with the Martini 3 FF could efficiently provide researchers with reasonable starting structures for these large complexes, whose dynamics can be probed by MD. This is the premise of the currently in-development CG version of LightDock (Roel-Touris *et al.*, 2020a, 2020b), implementing the Martini 3 FF. Coupling these tools with CG docking and MD simulations would allow to derive rules for PROTAC linker design and screening and/or to evaluate ternary complex stability when varying the Ligase protein. Within the field of drug delivery, the Martini 3 model combined with the implementation of tools and protocols available for constructing and simulating soft delivery systems, such as LNPs, will enable *in silico* screening of various formulations, permitting more efficient optimisation or rational design of delivery methods. However, for NA-containing drug delivery systems, the parameters for RNA/DNA are still under development in Martini 3 and the lack of experimentally resolved structures complicates FF parameter optimisation. While previous Martini 2 NA models were rather rigid (Uusitalo *et al.*, 2015, 2017), improving the dynamics of the future NA Martini 3 models is of utmost importance for the simulations of NA delivery systems.

Overcoming these challenges is fundamental for broader applications of the Martini 3 model in biologically relevant systems like SNs, protein–protein interactions, membrane systems and efficient discovery of druggable cryptic pockets, enabling an even larger impact of CG models in fields of drug discovery and delivery. For validation of the CG modelling within drug discovery, one example is the technique of co-crystallisation or soaking macromolecular crystals, essentially replacing solvent with a ligand within the crystal, enabling the comparison to for example, flooding CG-MD simulations for pocket identification (Wienen-Schmidt *et al.*, 2021). Within the drug delivery field, one could imagine correlating predicted structures and organisation of SNs based on CG-MD simulation with fusion and transfection efficacy (Miao *et al.*, 2020) measured experimentally, combined with fluorescence studies (Chen *et al.*, 2019) enhancing the understanding and development of such delivery methods.

**Open peer review.** To view the open peer review materials for this article, please visit http://doi.org/10.1017/qrd.2022.16.

**Financial support.** BJG is employed by Zymvol Biomodeling on a project which received funding from the European Union's Horizon 2020 research and innovation programme under Marie Skłodowska-Curie grant agreement No. 801342 (Tecniospring INDUSTRY) and the Government of Catalonia's Agency for Business Competitiveness (ACCIÓ). AB and MC received funding from the French National Research Agency (Grant no. ANR-18-CE11–0015). LM is supported by the French National Institute of Health and Medical Research (INSERM). PCTS, JM, GPP, and LRK are supported by the French National Center for Scientific Research (CNRS). Further funding of LRK, GPP, PCTS and LM came from a research collaboration with PharmCADD. SA and GR acknowledge the grant from the Interdisciplinary Centre for Clinical Research within the faculty of Medicine at the RWTH Aachen University (IZKF TN1-1/IA 532001; TN1–4/IA 532004) and the Deutsche Forschungsgemeinschaft (DFG) via the Research Training Group RTG2416 MultiSenses-MultiScales (368482240/GRK2416). AM and GR acknowledge the Helmholtz European Partnering fundings for the project 'Innovative high-performance computing approaches for molecular neuromedicine'. GR acknowledges the Federal Ministry of Education and Research (BMBF) and the state of North Rhine-Westphalia as part of the NHR Program, as well as the Joint Lab 'Supercomputing and Modeling for the Human Brain' of the Helmholtz Association, Germany and the two European Union's Horizon 2020 MSCA Program under grant agreement 956314 [ALLODD].

**Conflicts of interest.** The authors declare no conflicts of interest.

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
