## [Reviewer Report]

*Comments to Author*: This is an excellently written review of the current state-of-the art methods of using Martini 3 to study drugs and potential delivery systems. I found it to be an informative and interesting read, and have no suggestions at this stage to add to improve quality of the manuscript.

---

## [Reviewer Report]

*Comments to Author*: Kjølbye and co-authors present a very interesting and instructive review on the potential use of the MARTINI (mostly 3) force field to develop new research strategies for drug design and delivering. It is a well written manuscript with numerous recent and useful references. I would have only few comments to help the readership to contextualize the potential of such developments.

1- Throughout the manuscript (and especially in the introduction), it is not always clear what is done with MARTINI2 or MARTINI3. I understand that the authors want to promote MARTINI3 but this reviewer think that MARTINI2 was already useful to get qualitative results. Thus, it would be nice to precise a little bit more which MARTINI version was used.

2- Beyond MARTINI force field, atomistic simulations are still a gold standard to assess protein-ligand interactions and teams like DE Shaw showed recent successes (https://www.prnewswire.com/news-releases/d-e-shaw-research-licenses-first-in-class-therapeutic-for-immunological-diseases-to-lilly-301566618.html).So, it would be useful to balance a bit more the introduction by presenting recent successes from atomistic simulations.

3- In this review the authors present several examples to develop new drugs but seem to mainly focus on proteins or peptides. Would it be possible to highlight/discuss a bit more the development of this force field for small molecules ?For example, what is the position of the authors for the use of MARTINI force field for fragment based screening (like https://pubmed.ncbi.nlm.nih.gov/31030650/) ?

4- In section A- Protein conformation and cryptic pockets, the authors focused a lot on how MARTINI3 may help to highlight cryptic pockets but then how to use the protein structures ? Would it be used in combination with MARTINI3 small molecules - and in this case how the small molecules can find the pocket ? Would it be then backmapped into atomistic system and used for classical docking ?

5- In section C- Drugs targeting protein-protein interactions, the author mentioned the role of the flexibility for the linkers but did not seem tp present/discuss too much the work done with MARTINI forcefield on Intrinsically Disordered Proteins (IDPs) which may complement the references already presented in this section.

6- Overall, the authors present what it will be possible to do with MARTINI3 in term of modeling but do not present how it can then be compared or validated through experiments. Thus, for a broader audience, this information may help to know how to use MARTINI force field to answer specific experimental questions.

---

## [Reviewer Report]

*Comments to Author*: The perspective article by Kjølbye et al represents a fresh reading of the recent Martini 3 force field for different applications. The authors address several of the current problem in the methodology and I do find quite good the whole article. However, I suggest to include in the introduction a broad description of other methodologies which are concurrent to Martini 3.

In this way it will also to stress the robustness of Martini 3 as one of the advanced tool in biomolecular modelling. Other CG methodologies have shown relative success in the modelling of nucleic acids (OxDNA/OxRNA), as well as proteins (UNRES) and even implemented for drug discovery (CABS) and so on.

Line 189: Please be more specific about what author means by flavors, I guess it refers to different choices of parameters. Describe the parameters and tuning.

Section A, I believe the reader will be benefited by knowing what changes are present in the new Go-Martini with Martini 3 that is not present in the Poma et al JCTC 2017 original work.

Since the original Go-Martini was also parametrized on the basis of AFM data for nanomechanical studies. I brief description of the current nanomechanics studies could be included.

Authors show the relevance of the contact map in the parametrization of the Go-Martini model. A brief comment on the type of schemes such as atomic overlaps, chemical base or any other one which can assist the construction of a protein model should be mentioned.

In regard of the NA potential in Martini 3, I wonder whether the Uusitalo work using EN in Martini 2 will be revised. EN in protein is fundamental to keep secondary structure, in case of NA, the use of EN represent a very poor description of the stability of the NA (e.g. double helix which is a primary structure in NA). This means in the NA with Martini 3 one will expect to remove EN by improving the energetic description. Can the author comment on limitations of the Martini 2 respect to Martini 3 in regard of NA.

---

## [Reviewer Report]

*Comments to Author*: Reviewer #1: This is an excellently written review of the current state-of-the art methods of using Martini 3 to study drugs and potential delivery systems. I found it to be an informative and interesting read, and have no suggestions at this stage to add to improve quality of the manuscript.

Reviewer #2: The perspective article by Kjølbye et al represents a fresh reading of the recent Martini 3 force field for different applications. The authors address several of the current problem in the methodology and I do find quite good the whole article. However, I suggest to include in the introduction a broad description of other methodologies which are concurrent to Martini 3.

In this way it will also to stress the robustness of Martini 3 as one of the advanced tool in biomolecular modelling. Other CG methodologies have shown relative success in the modelling of nucleic acids (OxDNA/OxRNA), as well as proteins (UNRES) and even implemented for drug discovery (CABS) and so on.

Line 189: Please be more specific about what author means by flavors, I guess it refers to different choices of parameters. Describe the parameters and tuning.

Section A, I believe the reader will be benefited by knowing what changes are present in the new Go-Martini with Martini 3 that is not present in the Poma et al JCTC 2017 original work.

Since the original Go-Martini was also parametrized on the basis of AFM data for nanomechanical studies. I brief description of the current nanomechanics studies could be included.

Authors show the relevance of the contact map in the parametrization of the Go-Martini model. A brief comment on the type of schemes such as atomic overlaps, chemical base or any other one which can assist the construction of a protein model should be mentioned.

In regard of the NA potential in Martini 3, I wonder whether the Uusitalo work using EN in Martini 2 will be revised. EN in protein is fundamental to keep secondary structure, in case of NA, the use of EN represent a very poor description of the stability of the NA (e.g. double helix which is a primary structure in NA). This means in the NA with Martini 3 one will expect to remove EN by improving the energetic description. Can the author comment on limitations of the Martini 2 respect to Martini 3 in regard of NA.

Reviewer #3: Kjølbye and co-authors present a very interesting and instructive review on the potential use of the MARTINI (mostly 3) force field to develop new research strategies for drug design and delivering. It is a well written manuscript with numerous recent and useful references. I would have only few comments to help the readership to contextualize the potential of such developments.

1- Throughout the manuscript (and especially in the introduction), it is not always clear what is done with MARTINI2 or MARTINI3. I understand that the authors want to promote MARTINI3 but this reviewer think that MARTINI2 was already useful to get qualitative results. Thus, it would be nice to precise a little bit more which MARTINI version was used.

2- Beyond MARTINI force field, atomistic simulations are still a gold standard to assess protein-ligand interactions and teams like DE Shaw showed recent successes (https://www.prnewswire.com/news-releases/d-e-shaw-research-licenses-first-in-class-therapeutic-for-immunological-diseases-to-lilly-301566618.html).So, it would be useful to balance a bit more the introduction by presenting recent successes from atomistic simulations.

3- In this review the authors present several examples to develop new drugs but seem to mainly focus on proteins or peptides. Would it be possible to highlight/discuss a bit more the development of this force field for small molecules ?For example, what is the position of the authors for the use of MARTINI force field for fragment based screening (like https://pubmed.ncbi.nlm.nih.gov/31030650/) ?

4- In section A- Protein conformation and cryptic pockets, the authors focused a lot on how MARTINI3 may help to highlight cryptic pockets but then how to use the protein structures ? Would it be used in combination with MARTINI3 small molecules - and in this case how the small molecules can find the pocket ? Would it be then backmapped into atomistic system and used for classical docking ?

5- In section C- Drugs targeting protein-protein interactions, the author mentioned the role of the flexibility for the linkers but did not seem tp present/discuss too much the work done with MARTINI forcefield on Intrinsically Disordered Proteins (IDPs) which may complement the references already presented in this section.

6- Overall, the authors present what it will be possible to do with MARTINI3 in term of modeling but do not present how it can then be compared or validated through experiments. Thus, for a broader audience, this information may help to know how to use MARTINI force field to answer specific experimental questions.